# Genetic Structure of Invasive Baby’s Breath (*Gypsophila paniculata* L.) Populations in a Michigan Dune System

**DOI:** 10.3390/plants9091123

**Published:** 2020-08-31

**Authors:** Hailee B. Leimbach-Maus, Eric M. McCluskey, Alexandra Locher, Syndell R. Parks, Charlyn G. Partridge

**Affiliations:** 1Annis Water Resources Institute, Grand Valley State University (AWRI-GVSU), Muskegon, MI 49441, USA; hailee.leimbachmaus@gmail.com (H.B.L.-M.); syndell.parks@gmail.com (S.R.P.); 2Department of Biology, Grand Valley State University, Allendale, MI 49401, USA; mccluske@gvsu.edu (E.M.M.); lochera@gvsu.edu (A.L.)

**Keywords:** coastal sand dunes, invasive species, genetic diversity, landscape genetics, population structure

## Abstract

Coastal sand dunes are dynamic ecosystems with elevated levels of disturbance and are highly susceptible to plant invasions. One invasive plant that is of concern to the Great Lakes system is *Gypsophila paniculata* L. (perennial baby’s breath). The presence of *G. paniculata* negatively impacts native species and has the potential to alter ecosystem dynamics. Our research goals were to (1) estimate the genetic structure of invasive *G. paniculata* along the Michigan dune system and (2) identify landscape features that influence gene flow in this area. We analyzed 12 populations at 14 nuclear and two chloroplast microsatellite loci. We found strong genetic structure among populations (global F_ST_ = 0.228), and pairwise comparisons among all populations yielded significant F_ST_ values. Results from clustering analysis via STRUCTURE and discriminant analysis of principal components (DAPC) suggest two main genetic clusters that are separated by the Leelanau Peninsula, and this is supported by the distribution of chloroplast haplotypes. Land cover and topography better explained pairwise genetic distances than geographic distance alone, suggesting that these factors influence the genetic distribution of populations within the dunes system. Together, these data aid in our understanding of how invasive populations move through the dune landscape, providing valuable information for managing the spread of this species.

## 1. Introduction

Coastal sand dunes are dynamic ecosystems. Both the topography and biological community are shaped by disturbance from fluctuations in water levels, weather patterns, and storm events [1,2,3]. In these primary successional systems, vegetation plays an imperative role in trapping sand and soil, both of which accumulate over time and result in sand stabilization and dune formation [4,5,6]. Much of the vegetative community native to coastal dune systems is adapted to the harsh conditions posed by the adjacent coast, and some species require early successional, open habitat to thrive [2,7]. For example, dune species such as Marram grass (*Ammophila brevigulata* Fern.), Lake Huron tansy (*Tanacetum huronense* Nutt.), and Pitcher’s thistle (*Cirsium pitcheri* Torr. ex Eaton (Torr. & Grey)) are adapted to sand burial and will continue to grow above the sand height as it accumulates [7]. It is the heterogeneous topography and successional processes due to continuous disturbance that makes dune systems so unique [2].

Because coastal dune ecosystems have naturally elevated levels of disturbance, they are highly susceptible to plant invasions [8,9,10]. Invasive plant species are known to be adept at colonizing disturbed areas, and in sparsely vegetated dune systems that are often in early stages of succession, the opportunities for invasive colonizers are great [4,11,12,13]. Coastal dune systems also typically have a gradient of increasing stages of succession [14], and this heterogeneous structure can further promote various stages of an invasion, such as colonization, dispersal, and range expansion [15,16].

Within the Michigan dunes system, these successional processes have resulted in a patchwork pattern with alternating areas of open dune habitat, interdunal swales, shrub-scrub, and forested pockets scattered across the landscape [3,4,7]. This landscape structure can play an important role in shaping species migration, invasive spread, and population demographics [8,15,16], thus potentially driving patterns of population structure for invasive species.

In addition to the landscape, demographic processes during a species’ invasion also shape the genetic structure observed in contemporary populations. Multiple separate introduction events can result in contemporary populations that are genetically distinct from one another and from the native range [17,18,19]. Bottleneck events during an introduction can further limit the genetic variation in the invasive range, though this has not necessarily been found to limit the success of an invader [17,20]. Additionally, genetic admixture and inbreeding can shape the structure of populations, and the effect of these processes can be further influenced by the landscape structure and habitat heterogeneity [18,21,22,23].

*Gypsophila paniculata* L. (perennial baby’s breath, Caryophyllaceae) is an invasive plant that has been identified as a species of concern due to its impact on the integrity of the Michigan dune system [24]. A perennial iteroparous forb, *G. paniculata* is native to the Eurasian steppe region and was introduced to North American in the late 1880s [25,26]. Since its introduction, it has spread throughout the western United States and Canada and has established across a diverse array of habitats, including sand dunes, prairies, disturbed roadsides, and sage brush steppes [26,27,28]. In Michigan, *G. paniculata* negatively impacts the coastal dune community by crowding out sensitive species such as Pitcher’s thistle (*Cirsium pitcher*) through direct competition for limited resources, forming monotypic stands in the open dune habitat, preventing the reestablishment of native species, and limiting pollinator visits to native species [29,30,31]. *Gypsophila paniculata* dispersal is thought to be primarily wind-driven [26], which is also the mechanism that shapes the dunes. Following seed maturity, the stems of *G. paniculata* individuals become dry and brittle, breaking at the caudex and forming tumbleweed masses that can disperse roughly 10,000 seeds per plant up to 1 km [26,27]. Due to the topography and the heterogeneous habitat of the dune system, the wind patterns of this landscape have the potential to shape the structure of invasive *G. paniculata* populations. Wind can drive the direction and distance that tumbleweeds disperse, and it is possible that wind patterns could both promote gene flow or limit it by driving tumbleweeds into undesirable habitat (such as forested areas). Additionally, the steep topography in parts of the dunes could further prevent tumbleweeds from dispersing significant distances. With these interactive processes in mind, this study explored the genetic structure of invasive populations of *G. paniculata* within the Michigan coastal dune system.

Similar work assessing the population structure of invasive *G. paniculata* populations within the mid-western and western United States have found that these invasive populations are distributed among at least two distinct genetic clusters, both of which are present in the Michigan sand dunes [28]. Our goal for this study was to more extensively sample populations throughout Michigan to gain a better understanding of how these populations are distributed and how they disperse throughout the dune landscape. Thus, our specific aims were to (1) estimate the genetic structure of invasive populations at a finer geographic scale by focusing only on populations distributed throughout the Michigan dunes system and (2) identify landscape features within the dune system that hinder gene flow of *G. paniculata*. By estimating the structure of these invasive populations and the landscape features influencing their structure, we can better understand the impact the landscape and its dynamic processes have on this plant invasion, potentially leading to more effective management practices.

## 2. Results

### 2.1. Microsatellite Genotyping and Genetic Diversity

We genotyped 313 individuals from 12 locations along northwestern Michigan (Figure 1, Appendix A) at 14 nuclear microsatellite loci (nSSR) and two chloroplast microsatellite loci (cpSSR) (Appendix A). For the nSSR markers, no loci showed evidence of null alleles across all populations, there were no loci with more than four populations significantly out of Hardy–Weinberg Equilibrium (HWE) (less than 30% of populations) (Appendix A), and no loci significantly deviated from linkage equilibrium across all populations. The nSSR loci were moderately polymorphic, and the number of alleles per locus per population ranged from 1–11, with a total of 85 alleles across the 14 loci (Appendix A). Allelic richness (A_R_) ranged from 2.32–4.21 per population, and GM, PS, and TC populations exhibited significantly lower levels of A_R_ than the other populations (df = 154, t-value = 8.096, *p* = 1.61 × 10^−13^) (Table 1). Of the six private nSSR alleles identified, five were at low frequencies—occurring in five or fewer individuals—but the private nSSR allele in the GM population occurred in over 60% of individuals. Overall, the observed heterozygosity (H_O_) and expected heterozygosity (H_E_) tended to be lower in the three northernmost populations (GM, PS, TC) (Table 1). GM had a higher inbreeding coefficient (Table 1), but this could be attributed to our limited area in which to sample.

Both cpSSR loci were polymorphic, with three alleles per locus for a total of six alleles, and the number of alleles per population ranged from 2–4 (Appendix A). Haplotype richness ranged from 0.00–2.00 and haploid diversity ranged from 0.00–0.58 per population (Appendix A). All alleles together resulted in five haplotypes, and there were between 1–3 haplotypes per population (Table 1). One allele and haplotype 2 were both unique to the SB and ZP sampling locations, and another allele and haplotype 3 were both private to five individuals sampled in GM (Appendix A). These five individuals were collected from a separate sampling location from the rest of the individuals in GM.

### 2.2. Genetic Structure

The nSSR data suggested that there is strong genetic structure among the populations and regions of *G. paniculata* sampled along the dunes of Michigan (global F_ST_ = 0.23). Pairwise comparisons using the nSSR data among all 12 populations yielded significant F_ST_ values (Table 2). However, all pairwise comparisons of populations within Sleeping Bear Dunes National Lakeshore (hereafter Sleeping Bear Dunes or SBDNL) (GHB, SBP, DC, DP, EB, PB, SB) and nearby ZP displayed relatively lower pairwise F_ST_ values (Table 2). The AMOVA based on the nSSR data also found that a significant amount of the genetic variation could be explained by differences between populations primarily northeast of the Leelanau Peninsula (GM, PS, TC), populations southwest of the Leelanau Peninsula (GHB, SBP, DC, DP, EB, PB, SB, ZP, AD) (F_CT_ = 0.14, *p* < 0.0001) (Figure 1), and among populations within regions (F_SC_ = 0.10, *p* < 0.0001).

The Bayesian clustering analysis from the program STRUCTURE [34] partitioned the populations into two clusters (K = 2) (Figure 2A), inferred from Evanno’s ∆K [35] (Appendix A). This analysis was run without inferring any prior information on sampling location and then again with sampling information as prior. No differences were observed between the two results (without priors shown in Figure 2A). Cluster 1 is comprised of populations primarily northeast of the Leelanau Peninsula (GM, PS, TC), and cluster 2 includes populations southwest of the Leelanau Peninsula. However, five individuals in the GM population were assigned to cluster 2 (assignment probability >90%), and these individuals were located at a separate sampling location from the rest in GM. Though there was little admixture overall, several individuals in the GM, TC, EB, and AD populations showed a higher proportion of admixture among the two clusters.

A discriminant analysis of principal components (DAPC) scatterplot (Figure 2B) grouped individuals into three clusters along two axes, with AD separating from other individuals assigned to cluster 2. However, Figure 2C shows a large amount of overlap between the distributions of individuals in DAPC clusters 2 and 3 along the first discriminant function, suggesting little distance between them. The membership of individuals of each population to the three illustrated clusters can be seen in Figure 2D. These data further highlight the potential for subtle sub-structuring of *G. paniculata* populations in the dune system of Michigan but supports the STRUCTURE analysis of two main genetic clusters within the region.

A Mantel test for isolation by distance (IBD) performed over all populations found a significant correlation between genetic and geographic distances (R = 0.755, *p* < 0.001) (Appendix A). Upon further exploration of this correlation through separate Mantel tests within each identified STRUCTURE cluster, we found a significant correlation within cluster 2 (Appendix A) (R = 0.523, *p* = 0.030) but no significant correlation within cluster 1 (Appendix A) (R = 0.205, *p* = 0.494).

The AMOVA based on Φ_ST_ distance facilitated the comparison between the nSSR and cpSSR data, which resulted in a significant amount of the genetic variation explained by differences among regions (Φ_CT_), among populations within regions (Φ_SC_), and among all populations (Φ_ST_) for both data sets (*p* < 0.0001). Both datasets also showed that most of the variation was explained by among population differences (nSSR Φ_ST_ = 0.355, cpSSR Φ_ST_ = 0.736, *p* < 0.0001) (Appendix A).

For the cpSSR markers, the minimum spanning network illustrates the distribution of haplotypes across the 12 populations (Appendix A). Five haplotypes were found; Haplotype 1 was the most common, but only occurred in the SBDNL (GHB, SBP, DC, DP, EB, PB, SB) and ZP populations. Haplotype 2 was private to the SB and ZP populations, but rare, occurring in one and two individuals, respectively. Haplotype 3 was private to five GM individuals located separately from the majority of the other individuals from the GM population. Haplotype 4 was private to SB, ZP, and AD populations and occurred in all AD individuals, but it was less common in the SB and ZP populations. Haplotype 5 was private to GM, PS, and TC populations.

### 2.3. Landscape Genetics

Our landscape genetics analysis included nine of the 12 sampled populations, which corresponded to individuals grouped into cluster 2 based on the STRUCTURE analysis. We specifically chose to focus on these populations as they inhabit a continuous portion of the Michigan dune complex, allowing us to gain a better understanding of how dune morphology impacts the distribution of these populations. This analysis showed most single and combination surfaces (including surface area ratio, site exposure, topographic position index (TPI), and land cover) improved on geographic distance in explaining our genetic distance data based on marginal R^2^ and log-likelihood (Table 3). Model ranking with AIC_c_ selected the combination of all derived elevation surfaces (surface area ratio, site exposure, TPI) as the top model (Table 3). The surface area ratio contributed significantly more (86%) to the optimized combination elevation surface than site exposure (7%) or TPI (7%). Land cover has the highest R^2^ (marginal = 0.52 and conditional = 0.77) and log-likelihood (88.1) of all evaluated surfaces and is slightly favored by AIC model ranking. The beach/dune class had the lowest resistance value (1.0) of the four land-cover thematic classes, followed by forest (10.5), grass/shrub (945.9), and open water (5258.5). A genetic connectivity map of optimized resistance surfaces from land cover and the combination derived elevation surfaces show variation in locations of optimized gene flow (Figure 3).

## 3. Discussion

The natural disturbance regime of dynamic sand dune systems can result in a pattern of fragmented habitat and often sparse vegetative cover, making dune ecosystems highly susceptible to plant invasions [8,9,10]. The topography, geographic distribution of preferred habitat, and disturbance regime in an ecosystem can influence various stages of a species invasion, including where the plant establishes, its dispersal patterns, and how densely it grows [15,16]. In addition, the demographic processes of an introduction event can shape contemporary population dynamics [17,36]. The invasion of *G. paniculata* in the Michigan dune system is an opportunity to better understand the genetic structure of an invasive species in this system and how the dynamic landscape of these dunes may be shaping it.

Our results indicate that Michigan populations show strong genetic structure, and these populations primarily cluster into two distinct genetic groups that are separated by the Leelanau Peninsula. Populations in the cluster northeast of the Leelanau Peninsula (cluster 1: Traverse City, Petoskey State Park, and Grand Marais) tended to have lower levels of genetic diversity, based on effective allele number and observed heterozygosity, than those populations that are southwest of the Leelanau Peninsula (cluster 2: Sleeping Bear Dunes populations, Zetterberg Preserve, and Arcadia Dunes). These differences in genetic diversity could be due to a combination of invasion history and population demographic processes after introduction. The level of genetic diversity in the northeast cluster is similar to what has been observed in other invasive *G. paniculata* populations throughout the midwest and western US that also cluster with the Petoskey, MI population [28]. Invasion curves created through herbaria collections suggest that the establishment of these two genetic clusters may have occurred in two separate invasion events, with populations from cluster 1 establishing earlier (1910–1930s) than populations from cluster 2 (1940s) [28]. If these clusters were the result of separate invasion events, differences in the level of standing genetic variation during those early invasion periods may be contributing to the different levels we observe in these contemporary populations [28]. Differences in the level of genetic diversity among these regions in Michigan could also be due to differences in population size, as smaller populations could be more affected by the impact of genetic drift and potential inbreeding resulting in the observed lower levels of genetic diversity [37,38,39]. Sleeping Bear Dunes is a largescale infestation and has some of the highest densities of *G. paniculata* found within the Michigan coastal dunes, consisting of 50–75% of the vegetative cover and covering hundreds of acres in some areas [40,41] (E. Rice, unpublished data). The Grand Marais, Petoskey State Park, and Traverse City populations are smaller than those found in Sleeping Bear Dunes, with continuous populations often limited to less than 45 acres [42]. In addition, the level of geographic isolation between Grand Marais, Petoskey State Park, and Traverse City could also be contributing to the lower levels of genetic diversity observed in these areas. Grand Marais is located in Michigan’s upper peninsula, while Petoskey State Park and Traverse City are located in the lower peninsula. Traverse City, Petoskey State Park, and Grand Marais also have more human development along the lakeshore, which may provide additional barriers to gene flow among these populations. On the other hand, Sleeping Bear Dunes, Zetterberg Preserve, and Arcadia Dunes make up a large contiguous amount of land that has been preserved by the National Parks Service, The Nature Conservancy, and the Grand Traverse Regional Land Conservancy. In these areas, the dune habitat is often continuous, with limited human development. As our landscape analysis shows, extensive areas of expected high gene flow are visible in both the land cover and combination elevation resistance surfaces between these sites, and this likely helps to maintain higher levels of genetic diversity throughout this region.

While two main genetic clusters of *G. paniculata* occur in Michigan, we also identified more subtle population structuring of *G. paniculata* throughout the dune landscape, particularly for populations in the southwest cluster. First, our DAPC analysis suggests that the Arcadia Dunes population may group into a potential third genetic cluster. Variation in allele frequencies and decreased allelic richness are two factors that could explain the divergence of the Arcadia Dunes population, but there are no private alleles or other obvious patterns causing this population to cluster separately from nearby populations of Zetterberg Preserve and South Boundary in Sleeping Bear Dunes. The higher rates of admixture between the two main clusters observed in Arcadia Dunes compared to other populations could also be a reason for its divergence. However, what is driving this higher level of admixture in the Arcadia Dunes population is currently unknown.

Second, population pair-wise F_ST_ values among all of our sampling sites were significant, suggesting limits to continuous gene flow even among geographically close populations located in continuous dune habitat. Given the high potential of seed dispersal for *G. paniculata*, the moderate levels of divergence observed between some of these populations, particularly those located within Sleeping Bear Dunes, was unexpected. Other dune species, such as Pitcher’s thistle, show limited genetic differentiation among populations within Sleeping Bear Dunes (F_ST_ = 0.01–0.04) [43] despite the fact there is limited natural seed dispersal for this species [44]. However, certain populations of *G. paniculata* in Sleeping Bear Dunes exhibited F_ST_ values between 0.068 - 0.1 when compared to other populations in the same area. This degree of differentiation was specifically observed for our Empire Bluff population which is located on the tip of a dune bluff—a small visitor outlook point surrounded by forest—and seems to be isolated from nearby populations. This suggests that some of the landscape features associated with the dunes system are contributing to this lack of continuous gene flow.

Within Sleeping Bear Dunes and the surrounding area, we found that population structure was significantly associated with a combination of geographic distance, dune topography, and habitat heterogeneity of the dune system. IBD revealed a moderate positive relationship between nSSR genetic distances and geographic distances across all of the sampled populations, and this positive relationship was also found for the analysis of our southwestern cluster. Both land cover and dune topography (surface area ratio, site exposure, topographic position index) showed an increase in improvement toward explaining the genetic distance data associated with these populations. *Gypsophila paniculata* is typically found in open back dune habitat but has also been found in the fore dunes close to the lake beach and on steep dune sides. Out of our characterized land cover features, the open beach/dune habitat showed the lowest resistance to gene flow, and these areas likely serve as corridors allowing *G. paniculata* to spread. Forested areas showed the second lowest level of resistance, which is contrary to what we initially expected. Forested areas are part of the back dunes and have been suggested by land managers as barriers between populations (personal communication, Shaun Howard and Jon Throop). The reason for this incongruity with on the ground observations is likely due to the resolution of the land cover surface. At 45-m resolution, some areas with narrow strips of beach (see Figure 3) are still classified as forest, making forest appear to be less of a barrier. The forest class also included any areas with trees regardless of density, so some gene flow may persist through smaller forest patches in the dunes or where trees are more spaced apart. Thus, while geographic distance influences the strong structuring of distant populations, the isolating effect of the topography and land cover within the dunes could have an effect that overrides that of geographic distance on smaller spatial scales such as that observed in Sleeping Bear Dunes. Additional fine scale sampling of *G. paniculata* in this region will further elucidate the roles both land cover and topography play in facilitating gene flow.

The population structure of invasive plants within complex habitats may also be disproportionally influenced by different rates of seed dispersal and pollination. The tumbleweed mechanism of dispersal that *G. paniculata* employs could be an effective means to disperse seeds, but as previously stated, the topographical structure and habitat heterogeneity patterns within the dunes may impact this process more than they impact pollination. Tumbleweeds are often observed to collect at the edge of forested areas or at the base of larger dunes. However, *G. paniculata* has been found to attract a diverse array of pollinator species [29,31], sometimes at the expense of native plant pollination. The variation in Φ_ST_ values between the two marker types (nSSR Φ_ST_ = 0.355, cpSSR Φ_ST_ = 0.736) indicates that barriers to seed dispersal may be more limiting for gene flow than pollination. While seeds can be dispersed up to 1 km, Darwent [27] also suggested that many of the seeds are released near the parent plant prior to the stems tumbling, resulting in strong population structure due to a lower frequency of migrants. Therefore, the physical elements of the dune ecosystem could be impacting gene flow through seed dispersal by further limiting the plant’s ability to spread throughout the landscape (e.g., the presence of vegetation, open water). A similar notion has been expressed for other *Gypsophila* species inhabiting harsh habitats within their native range. For example, *G. struthium* Loefl. is endemic to Iberian gypsum outcrops, and the landscape of this ecosystem leads to fragmented and disjointed areas of suitable habitat. These landscape features can limit gene flow among populations, and, similar to our work, pollination seems to play a more dominant role in facilitating gene flow compared to seed dispersal [45]. However, these comparisons of cpSSR results to nSSR for our *G. paniculata* data should be taken with some caution, as we had a limited number of polymorphic cpSSR markers. Though we chose to use microsatellites within the chloroplast genome to increase the likelihood of polymorphisms, we still found these regions to be well-conserved and with limited variation in our dataset. Therefore, we cannot rule out the possibility of fragment size homoplasy confounding results of low genetic diversity in some populations [46].

One additional mechanism that could be contributing to the movement of *G. paniculata* throughout the dunes that we were not able to address is the role of human-mediated transport. Humans have been shown to play an accidental role in the spread of invasive species as seeds can become trapped in clothing and then moved to different locations [47,48]. The coastal Michigan dune system is a popular recreation area among locals and tourists. The autumn season brings a high volume of foot traffic, and it is possible that people may be accidentally transporting *G. paniculata* seeds between these otherwise isolated populations, as the seed phenology coincides with the autumn senescence. Given the small size of these seeds (~2 mm) and high seed abundance at these locations in the fall, visitors may be moving seeds from one area to another as seeds become caught in their clothing or shoes. This type of movement could help explain some of the higher levels of admixture found in some populations, such as Arcadia Dunes, or the distinct clustering of some individuals collected in Grand Marais. The five individuals from the Grand Marais population that clustered with the populations southwest of the Leelanau Peninsula were collected in a separate area from the other Grand Marais individuals. This area was directly adjacent to a public parking lot and public pier. These five individuals also had distinct chloroplast haplotypes that differed from the other Grand Marais individuals and all other populations we sampled. We suggest further studies should be conducted to evaluate how human-mediated transport of *G. paniculata* via seeds impacts the dispersal and spread of this invasive plant throughout the dunes.

The information gained from assessing the population structure of invasive species can help inform and improve management efforts designed to control and/or eradicate invasive plants. Given the strong clustering of *G. paniculata* among populations to the north and south of the Leelanau Peninsula, it may be beneficial for land managers to focus on treating areas at the intersection of these two clusters or to focus on limiting seed movement between these areas to decrease the potential for increased admixture (mixing of genetically distinct populations). The justification for this is that increased admixture can lead to increased genetic diversity or fitness through heterosis, potentially resulting in higher adaptive potential and reproductive output in these invasive populations [49,50,51]. Identifying how populations are connected throughout the landscape can also aid in more effective management plans. For example, populations within Sleeping Bear Dunes show higher to moderate levels of gene flow, suggesting seed dispersal from local populations is occurring. With these moderate levels of gene flow, it is important to realize that even if *G. paniculata* is completely removed from one of these areas, it is possible for it to become re-established due to seed dispersal from connected populations. Finally, identifying important landscape features within the dunes system that help promote or restrict gene flow can aid in identifying optimal areas in which new populations may spread and establish, thus helping future monitoring efforts.

## 4. Materials and Methods

### 4.1. Study Area and Sampling Collection

To determine the population structure on a regional scale in Michigan, we collected leaf tissue samples from plants at 12 different sites in the summers of 2016–2017. All sites were located in areas of known infestation along the dune system of Michigan (Figure 1, Appendix A), and the majority have a history of treatment primarily by The Nature Conservancy, the Grand Traverse Regional Land Conservancy, and the National Park Service. Eleven sites were located along Lake Michigan in the northwest lower peninsula of Michigan, and one was located on Lake Superior in the upper peninsula. We collected leaf tissue samples (5–10 leaves per individual) from a minimum of 20 individuals per site (maximum of 35 individuals) and stored the tissue in individual coin envelopes in silica gel until DNA extractions took place (total *n* = 313). Site locations in Michigan were separated by a minimum of 10 km and a maximum of 202 km.

### 4.2. Microsatellite Genotyping

We extracted genomic DNA from all samples using DNeasy plant mini kits (QIAGEN, Hilden, Germany) and followed supplier’s instructions with minor modifications, including an extra wash step with AW2 buffer. We then ran the extracted DNA twice through Zymo OneStep PCR Inhibitor Removal Columns (Zymo, Irvine, CA, USA) and quantified the concentrations on a Nanodrop 2000 (Thermo Fisher Scientific, Waltham, MA, USA). We included deionized water controls in each extraction as a quality control for contamination.

We amplified samples at 14 polymorphic nuclear microsatellite loci (hereafter nSSRs) that were developed specifically for *G. paniculata* (Appendix A) [52]. We conducted polymerase chain reactions (PCR) using a forward primer with a 5′-fluorescent labeled dye (6-FAM, VIC, NED, or PET) and an unlabeled reverse primer. PCRs consisted of 1× Taq buffer with KCl (Thermo Scientific, Waltham, MA, USA), 2.0–2.5 mM MgCl_2_ depending on the locus (Appendix A); 300 µM dNTP; 0.08 mg/mL bovine serum albumin (BSA); 0.4 µM forward primer fluorescently labeled with either FAM, VIC, NED, or PET; 0.4 µM reverse primer; 0.25 units of Taq polymerase (Thermo Scientific, Waltham, MA, USA); and a minimum of 50 ng DNA template, all in a 10.0 µL reaction volume [52]. The thermal cycle profile consisted of denaturation at 94 °C for 5 min followed by 35 cycles of 94 °C for 1 min, annealing at 62 °C for 1 min, extension at 72 °C for 1 min, and a final elongation step of 72 °C for 10 min.

Each sample was also amplified at two universal chloroplast microsatellite loci (hereafter cpSSRs) previously developed for *Nicotiana tabacum* L. [53] (ccssr4, ccssr9) (Appendix A). PCR reactions were conducted using a forward primer with a 5′-fluorescently labeled dye (NED or PET) (Applied Biosystems, Foster City, CA, USA) and an unlabeled reverse primer. PCR reactions for the cpSSRs are the same as detailed above for the nuclear loci. The thermal cycler profile for cpSSRs is as follows: denaturation at 94 °C for 5 min followed by 30 cycles of 94 °C for 1 min, annealing at 52 °C for 1 min, extension at 72 °C for 1 min, and a final elongation step of 72 °C for 8 min (modified from Calistri et al. [54]).

We determined successful amplification by visualizing the amplicons on a 2% agarose gel stained with ethidium bromide. For both nSSR and cpSSR markers, we multiplexed PCR amplicons according to dye color and allele size range (Appendix A), added LIZ Genescan 500 size standard, denatured with Hi-Di formamide at 94 °C for four minutes, and then performed fragment analysis on an ABI3130xl Genetic Analyzer (Applied Biosystems, Foster City, CA, USA). We genotyped individuals using the automatic binning procedure on Genemapper v5 (Applied Biosystems, Foster City, CA, USA) and constructed bins following the Genemapper default settings. To account for the risk of genotyping error when relying on an automated allele-calling procedure, we visually verified that all individuals at all loci were correctly binned to minimize errors caused by stuttering, low heterozygote peak height ratios, and split peaks [55,56].

### 4.3. Quality Control

Prior to any analysis, we used multiple approaches to check for scoring errors [55]. We checked nSSR genotypes for null alleles and potential scoring errors due to stuttering and large allele dropout using the software Micro-Checker v2.2.3 [57,58]. Prior to marker selection, the loci used in this study were previously checked for linkage disequilibrium [52]. We checked for heterozygote deficiencies in the package STRATAG in the R statistical program. We then screened our data for individuals with more than 20% missing loci and for loci with more than 10% missing individuals [59,60]. We found none, so all individuals and loci remained for further analyses. In addition, we genotyped 95 individuals twice to ensure consistent allele calls. For the nSSR dataset, we used Genepop 4.2 [61,62] to perform an exact test of HWE with 1000 batches of 1000 Markov Chain Monte Carlo iterations. We also checked for loci out of HWE in more than 60% of the populations; however, there were none.

### 4.4. nSSR Genetic Diversity

We calculated the total number of alleles per sampling location, private alleles, Shannon’s Identity Index, observed and expected heterozygosity, and estimated the inbreeding coefficient (F_IS_) in GenAlEx 6.502 [63,64]. We used the package diverSity in the R statistical program to calculate the allelic richness at each sampling location [65].

### 4.5. nSSR Genetic Structure

To test for genetic differentiation between all pairs of sampling locations, we calculated Weir and Cockerham’s [32] pairwise F_ST_ values for 9999 permutations in GenAlEx 6.502 [63,64]. In the R statistical program, we corrected the *p*-values using a false discovery rate (FDR) correction [33]. To test how much of the genetic variance can be explained by within and between population variation, we ran an analysis of molecular variance (AMOVA) for 9999 permutations in GenAlEx 6.502 [63,64].

To examine the number of genetic clusters among our sampling locations, we used the Bayesian clustering program STRUCTURE v2.3.2 [34]. Individuals were clustered assuming the admixture model both with and without a priori sampling locations for a burnin length of 100,000 before 1,000,000 repetitions of MCMC for 10 iterations at each value of *K* (1–16). The default settings were used for all other parameters. We identified the most likely value of *K* using the ∆K method from Evanno et al. [35] in CLUMPAK [66].

To further explore the genetic structure of these populations, we performed a Discriminant Analysis of Principal Components (DAPC) in the R package adegenet, which optimizes among-group variance and minimizes within-group variance [67,68]. To identify the number of clusters for the analysis, a clustering algorithm within the adegenet package was run for values of *K* clusters (1–16). We retained a *K*-value of 3 based upon the Bayesian information criteria (BIC) value. DAPC can be beneficial, as it can limit the number of principal components (PCs) used in the analysis. It has been shown that retaining too many PCs can lead to over-fitting and instability in the membership probabilities returned by the method [68]. Therefore, we performed the cross-validation function to identify the optimal number of PCs to retain. Out of 69 total PCs, the cross-validation function suggested we retain 60 PCs. We ran the DAPC using the recommended 60 PCs but also checked if the general patterns remained with fewer PCs used by running the analysis with incrementally less PCs (45 and 30 PCs). All general patterns of the data in the scatterplots remained consistent despite the decreased PCs; therefore, we chose to use the scatterplot based on 30 PCs, as the benefit of the DAPC for our purposes is to show that the main patterns remain, despite minimization of within population variation.

To assess the effect of isolation by distance (IBD), we used a paired Mantel test based on a distance matrix of Slatkin’s transformed F_ST_ (D = F_ST_/(1–F_ST_)) [69] and a geographic distance matrix for 9999 permutations in GenAlEx 6.502, and the analysis follows Smouse et al. [70] and Smouse and Long [71]. The mean geographic center was generated for each sampling location in ArcGIS software (ESRI^TM^ 10.4.1, Redlands, CA, USA), and the latitude and longitude of these points was then used to construct a matrix of straight-line distances in km between each sampling location. The reported *p*-values are based on a one-sided alternative hypothesis (H_1_: R > 0). A Mantel test was run for all sampling locations together, and a test was also run separately for populations within each cluster identified in the STRUCTURE analysis.

### 4.6. cpSSR Genetic Diversity

For the cpSSR dataset, we used the program HAPLOTYPE ANALYSIS v1.05 [72] to calculate the number of haplotypes, haplotype richness, private haplotypes, and haploid diversity for each population and microsatellite locus. To visualize patterns in the cpSSR dataset, we created a minimum spanning network in the R package poppr [73]. Nei’s genetic distance [74] based on individual haplotypes was used as the basis for the network with a random seed of 9999.

### 4.7. nSSR and cpSSR Genetic Structure

In order to compare the population structure of the nSSR and cpSSR data, we used the Φ_ST_ distance matrix for both datasets and ran an AMOVA. The population pairwise Φ_ST_ matrix facilitates comparison of molecular variance between codominant and dominant data by suppressing within individual variation, thus allowing for the comparison between varying mutation rates [32,75]. To test how much genetic variation could be explained by among populations, among populations within regions, and between regions (genetic clusters identified through STRUCTURE analysis) for both datasets, we ran an AMOVA for 9999 iterations in GenAlEx 6.502 [63,64].

### 4.8. Landscape Genetics

We conducted our analysis using genetic distance data (F_ST_) from nine of the 12 sites that corresponded to individuals belonging to the cluster southwest of the Leelanau Peninsula. The reason for this is because many of these populations are in closer geographic proximity to one another and within a continuous portion of the dunes system, allowing us to gain a better understanding of how natural dune landscape features impacted the genetic distribution of these populations. We used the R package ResistanceGA [76,77] to evaluate the influence of different landscape features on gene flow. ResistanceGA can assess both continuous and categorical data, optimizing resistance values for individual or combined surfaces based on pairwise genetic distances.

To determine landscape features within the dune system that hindered or facilitated gene flow, we initially classified a 75 × 5-km area along the Lake Michigan shoreline using high resolution (60 cm) imagery from the National Agriculture Inventory Program (2016) in ArcGIS v 10.4.1 (Environmental Systems Research Institute, Redlands, CA, USA). We used the maximum likelihood classification method in the Image Classification tool to classify the landscape into five categories: forest, water, sand, grass (i.e., beach grass), and open field (i.e., agriculture or old field). In addition to land cover, we used a 30-m digital elevation model (DEM) and ArcGIS Toolboxes [78,79] to generate derived elevation surfaces to assess dune topography. All the surfaces were initially run at resolutions of 30 m and 60 m. We implemented a two-step hierarchical protocol where the first step explored each candidate surface’s ability to explain genetic distance using ResistanceGA. We used the results from the first step to remove correlated surfaces, refine the land cover classification such that contiguous features were adequately represented (e.g., sand along the shoreline was still evident at all resolutions), and identify an appropriate resolution for developing a model that explained genetic distance. Ultimately, we included resistance surfaces for land cover (forest, open water, beach/dune, grasses/shrubs), surface area ratio [79], site exposure [79], and a topographic position index (TPI; [78]) at a 500 m scale to be assessed individually and in pairwise combinations with ResistanceGA. The final resistance surfaces were set at a resolution of 45-m, as a 30-m resolution was intractable, and a 60-m resolution was too coarse to capture contiguous linear landscape features. We used AIC and AIC_c_ for model selection of optimized surfaces and visualized genetic connectivity for these surfaces using Circuitscape [80].

## 5. Conclusions

The invasion of *G. paniculata* to the Great Lakes has the potential to disrupt the dynamism of the dune landscape and biological community in Michigan, and this threat has led to increased concerns over its pervasiveness regionally and nationally. Estimating the genetic structure of invasive populations can lead to a better understanding of the invasion history and the factors influencing the success of an invasion [18,81,82]. Through population level analysis, we found strong genetic structure present that separates the invasion in the Michigan dunes into two main regions. The genetic structure identified for these *G. paniculata* populations probably results from a combination of invasion history and demographic processes—isolation and admixture, along with landscape level processes. The topography of the dunes is heterogeneous but also constantly shifting, and the *G. panculata* invasion is one example of how this dynamic system can shape the establishment, gene flow, and spread of invasive plant populations.

## Figures and Tables

**Figure 1 plants-09-01123-f001:**
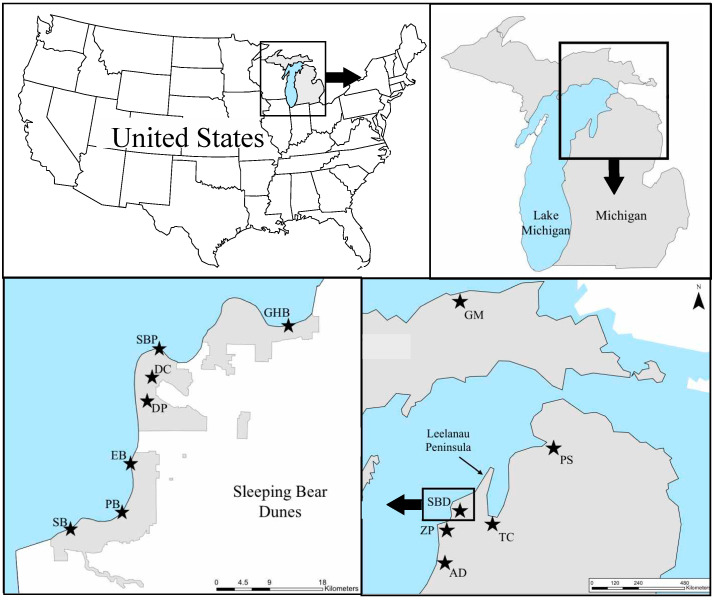
Map of *Gypsophila paniculata* sampling locations in Michigan. Seven were located throughout Sleeping Bear Dunes National Lakeshore. The park boundary is delineated by grey shading in bottom left panel. Sampling location codes: Grand Marais (GM), Petoskey State Park (PS), Traverse City (TC), Good Harbor Bay (GHB), Sleeping Bear Point (SBP), Dune Climb (DC), Dune Plateau (DP), Empire Bluffs (EB), Platte Bay (PB), South Boundary (SB), Zetterberg Preserve (ZP), Arcadia Dunes (AD).

**Figure 2 plants-09-01123-f002:**
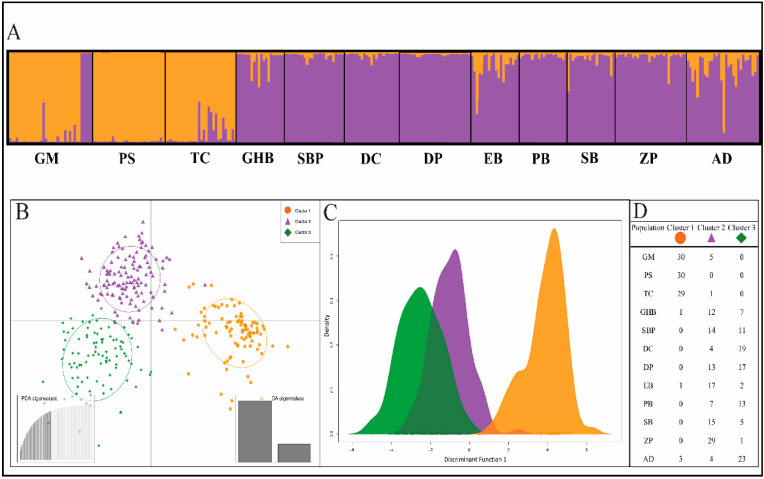
Identification of K clusters for *G. paniculata* in the MI dunes system. (**A**) Results from Bayesian cluster analysis based on nSSR data using the program STRUCTURE indicate (K = 2) population clusters. Cluster 1 (left, orange) includes the primarily northeast populations, and Cluster 2 (right, purple) includes all other populations, (**B**) Scatterplot of both discriminant function axes from the discriminant analysis of principal components (DAPC), (**C**) DAPC sample distribution on discriminant function 1, (**D**) Table of individual membership to each DAPC cluster, explained by the PCA eigenvalues used in the DAPC, based on all 69 identified principal components. Sampling location codes: Grand Marais (GM), Petoskey State Park (PS), Traverse City (TC), Good Harbor Bay (GHB), Sleeping Bear Point (SBP), Dune Climb (DC), Dune Plateau (DP), Empire Bluffs (EB), Platte Bay (PB), South Boundary (SB), Zetterberg Preserve (ZP), Arcadia Dunes (AD).

**Figure 3 plants-09-01123-f003:**
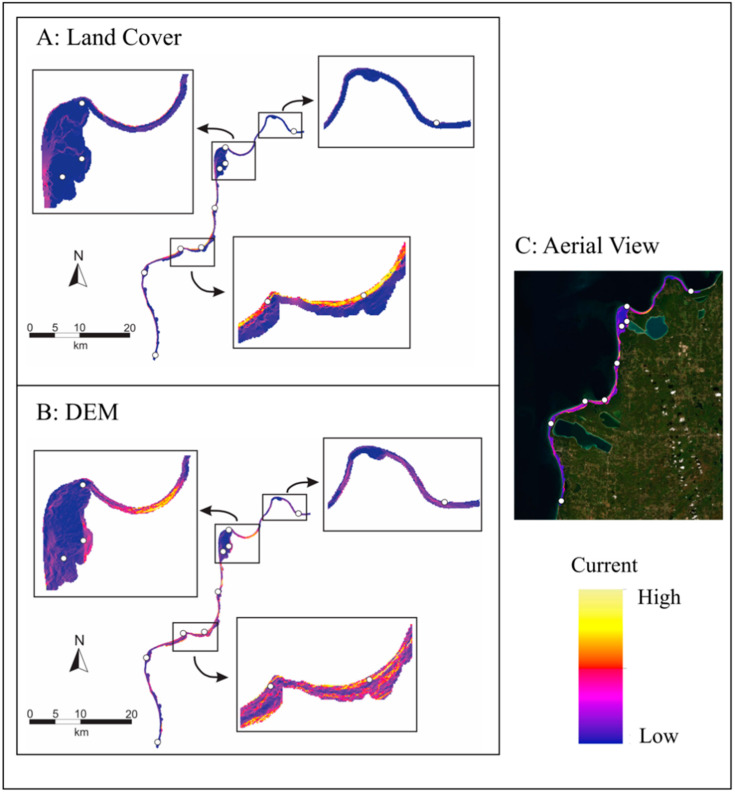
Genetic connectivity map for both (**A**) land cover and (**B**) digital elevation model-based (DEM) optimized resistance surfaces for populations belonging to cluster 2, southwest of the Leelanau Peninsula. Aerial view of the digitized portion is outlined in (**C**). White circles indicate sampling locations, and from north to south, they include Good Harbor Bay (GHB), Sleeping Bear Point (SBP), Dune Climb (DC), Dune Plateau (DP), Empire Bluffs (EB), Platte Bay (PB), South Boundary (SB), Zetterberg Preserve (ZP), and Arcadia Dunes (AD). The color indicates a current gradient, with dark purple indicating low current (high resistance) and yellow indicating high current (low resistance).

**Table 1 plants-09-01123-t001:** Genetic diversity indices for *G. paniculata* from each sampling location across 14 nuclear microsatellite loci (nSSRs) and two chloroplast microsatellite loci (cpSSRs). Values in parentheses represent standard error.

Sampling Locations												
	GM	PS	TC	GHB	SBP	DC	DP	EB	PB	SB	ZP	AD
N	35	30	30	20	25	23	30	20	20	20	30	30
**nSSR**												
H_O_	0.25 (0.06)	0.32 (0.07)	0.31 (0.05)	0.53 (0.06)	0.50 (0.05)	0.49 (0.06)	0.54 (0.05)	0.50 (0.07)	0.56 (0.06)	0.51 (0.05)	0.56 (0.05)	0.50 (0.5)
H_E_	0.30 (0.07)	0.33 (0.07)	0.33 (0.05)	0.56 (0.05)	0.55 (0.05)	0.53 (0.06)	0.57 (0.05)	0.05 (0.07)	0.56 (0.05)	0.52 (0.04)	0.55 (0.05)	0.54 (0.04)
F_IS_	0.14 (0.05)	0.02 (0.04)	0.03 (0.05)	0.05 (0.06)	0.09 (0.05)	0.08 (0.05)	0.04 (0.03)	−0.02 (0.04)	0.01 (0.02)	0.01 (0.04)	−0.05 (0.3)	0.06 (0.07)
I	0.56 (0.18)	0.55 (0.11)	0.58 (0.1)	1.07 (0.13)	1.02 (0.11)	1.01 (0.14)	1.07 (0.13)	0.96 (0.14)	1.06 (0.14)	1.02 (0.12)	1.05 (0.12)	0.92 (0.10)
% Poly Loci	85.71	85.7	92.9	100	100	100	100	92.86	100	100	100	100
A_R_	2.66	2.32	2.54	3.97	3.75	3.92	3.99	3.66	4.07	4.21	4.19	3.12
**cpSSR**												
N_H_	2 ^P^	1	1	1	1	1	1	1	1	3	3	1
H_R_	0.991	0	0	0	0	0	0	0	0	2	1.897	0

Notes: N—number of individuals, H_O_—observed heterozygosity, H_E_—expected heterozygosity, F_IS_—inbreeding coefficient, I—Shannon Identity Index, % Poly Loci—number of polymorphic loci per population, A_R_—allelic richness for each population averaged across loci, N_H_—number of haplotypes for each population averaged across loci, H_R_—haplotype richness for each population averaged across loci. ^P^ denotes a private haplotype. Sampling location codes: Grand Marais (GM), Petoskey State Park (PS), Traverse City (TC), Good Harbor Bay (GHB), Sleeping Bear Point (SBP), Dune Climb (DC), Dune Plateau (DP), Empire Bluffs (EB), Platte Bay (PB), South Boundary (SB), Zetterberg Preserve (ZP), Arcadia Dunes (AD).

**Table 2 plants-09-01123-t002:** Pairwise F_ST_ values for nSSR data among all sampling locations based on Weir and Cockerham’s [32] estimate. Darker color—increasing F_ST_ value, lighter color—decreasing F_ST_ value.

	GM	PS	TC	GHB	SBP	DC	DP	EB	PB	SB	ZP	AD
**GM**	−	−	−	−	−	−	−	−	−	−	−	−
**PS**	0.22	−	−	−	−	−	−	−	−	−	−	−
**TC**	0.15	0.12	−	−	−	−	−	−	−	−	−	−
**GHB**	0.26	0.25	0.22	−	−	−	−	−	−	−	−	−
**SBP**	0.26	0.25	0.23	0.05	−	−	−	−	−	−	−	−
**DC**	0.32	0.32	0.28	0.06	0.04	−	−	−	−	−	−	−
**DP**	0.27	0.27	0.25	0.03	0.03	0.03	−	−	−	−	−	−
**EB**	0.22	0.24	0.18	0.08	0.07	0.09	0.08	−	−	−	−	−
**PB**	0.29	0.28	0.26	0.06	0.04	0.05	0.05	0.11	−	−	−	−
**SB**	0.25	0.27	0.21	0.09	0.06	0.08	0.07	0.07	0.07	−	−	−
**ZP**	0.24	0.24	0.21	0.07	0.04	0.09	0.06	0.07	0.08	0.02	−	−
**AD**	0.30	0.25	0.23	0.12	0.12	0.14	0.12	0.17	0.13	0.13	0.13	−

Notes: All values significant at *p* ≤ 0.005 after false discovery rate (FDR) correction [33]. Sampling location codes: Grand Marais (GM), Petoskey State Park (PS), Traverse City (TC), Good Harbor Bay (GHB), Sleeping Bear Point (SBP), Dune Climb (DC), Dune Plateau (DP), Empire Bluffs (EB), Platte Bay (PB), South Boundary (SB), Zetterberg Preserve (ZP), Arcadia Dunes (AD).

**Table 3 plants-09-01123-t003:** Model ranking from ResistanceGA for land cover (LC), surface area ratio (SAR), site exposure (SE), and topographic position index (TPI).

Surface	k	AIC	AICc	R2m	R2c	LL	Delta.AICc	Weight
SAR, SE, TPI	10	−166.71	−264.71	0.28	0.68	87.36	0.00	1.00
LC, SAR, SE	11	−167.23	−241.23	0.30	0.68	87.62	23.48	0.00
LC, SAR, TPI	11	−166.75	−240.75	0.26	0.67	87.38	23.96	0.00
LC, SE, TPI	11	−166.04	−240.04	0.21	0.69	87.02	24.67	0.00
Distance	2	−164.88	−166.88	0.19	0.71	86.44	97.83	0.00
Null	1	−159.82	−163.25	0.00	0.67	82.91	101.46	0.00
SAR	4	−166.90	−156.90	0.29	0.68	87.45	107.82	0.00
SE	4	−166.66	−156.66	0.26	0.67	87.33	108.06	0.00
TPI	4	−166.03	−156.03	0.24	0.70	87.01	108.69	0.00
LC	5	−168.28	−146.28	0.52	0.77	88.14	118.44	0.00
SAR, SE	7	−166.82	−48.82	0.28	0.68	87.41	215.90	0.00
SAR, TPI	7	−166.82	−48.82	0.28	0.68	87.41	215.90	0.00
SE, TPI	7	−166.30	−48.30	0.24	0.68	87.15	216.41	0.00
LC, SAR	8	−167.49	Inf	0.31	0.68	87.74	Inf	0.00
LC, SE	8	−166.01	Inf	0.21	0.68	87.00	Inf	0.00
LC, TPI	8	−165.88	Inf	0.18	0.69	86.94	Inf	0.00

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
