# Peer review of "Genetic Structure of Invasive Baby’s Breath (Gypsophila paniculata L.) Populations in a Michigan Dune System"

_plants, 2020, doi:10.3390/plants9091123_

Round 1
Reviewer 1 Report
Overall, this is a well-written manuscript, with clear details of methods, results, and inferences. I do think the ms would be improved by having additional details of the broader geographical and historical context for this invasive species. For example, given that you found 2 genetic clusters, could these have resulted from separate introductions?
I have also made some suggestions for minor corrections in the manuscript file, and I note the following two edits in the supplementary materials:
1) Table S2 - Why do you include the sampling location codes when these are not included in the table itself?
2) Fig. S3 - In the legend change P to PS so it matches all other figures and tables.

Reviewer 2 Report
This is a good work because you have done a very good field surveys across the entire territory and statistical analyses even with Geographical Information Systems. Otherwise, I have a lot of recommendations to increase the quality of your paper. Be careful with the writing and mistakes.
The keywords “genetic structure” and “Gypsophila paniculata” are repeated in the article title. In order to increase the visibility of your paper I recommend changing these keywords. If you change them for same different keywords, you will increase the probability that your paper could be found by future readers when they look for your paper in some databases like Scopus for example. If you repeat the same words in the article title and in keywords, less people could find your work. So, you must think about the visibility of your research.
Please, put the keywords in alphabetical order. The journal publishes the keywords in this way. Follow the rules of the journal.
Just because you are going to publish this paper in a botanical journal the first time youuwrite a scientific name you must put the authors of them. So, you must put the authors of Gypsophila paniculate in the main title and in the abstract as well.
In the rest of the paper you must follow the same rule.
In line 37 you must put the authors to Ammophila brevigulata.
In line 38 you must put the authors of both plants Tanacetum huronense and Cirsium pitcheri.
In order to improve this very good paper I really encouraged you to read a very similar paper with another plant of the same genera (Gypsophila struthium) which grows in gypsum habitats, which are too very harsh habitats. This paper is the following one:
Martínez-Nieto, M.I., Segarra-Moragues, J.G., Merlo, E., Martínez-Hernández, F., Mota, J.F. Genetic diversity, genetic structure and phylogeography of the Iberian endemic Gypsophila struthium (Caryophyllaceae) as revealed by AFLP and plastid DNA sequences: Connecting habitat fragmentation and diversification. (2013) Botanical Journal of the Linnean Society, 173 (4), pp. 654-675.
Just in case you consider the information contained in this paper useful for your paper then you can freely cite this one if you precise it.
You must improve Figure 1. You have three maps and the first one, the left-upper one is not completed. If you look at this map you can not recognize where the water is following the natural colors, you have used. I propose to put another general map where you can see the limits of Canada, and United States and in this map to put the rectangle. Or maybe use four maps. Feel free to improve the Figure 1, but try to make a better map, and remember, two of the three maps are well done. As well I suggest you to use a general map of North America and then go to the place where you have done your study, maybe your left-upper map (with better colors, using the blue color for the water, and draw the limits between Canada and United States with a bold black line).
In line 167 you should write “Isolation By Distance” before “(IBD)”. I think it is better to write this way for future readers.
For the same reason in line 191 you should write “Topographic Position Index” just before “(TPI)”.
In line 374 you should write “Polymerase Chain Reactions” just before “(PCR)”. Review all the paper just in case you can find mistakes like these. Try to write as much clear as you can.
In line 428 you should write “Discriminant Analysis of Principal Components” just before “(DAPC)”.
For example, in line 432 “Bayesian Information Criteria (BIC)” is very well written, so try to do the same in all the paper.
In line 433 you should write “Principal Components” just before “(PCs)”.
Once more in line 442 you should write “Isolation By Distance” just before “(IBD)”.
In line 488 you should write “Topographic Position Index” just before “(TPI; [75])”.
